# Proposal of *Lentzea deserti* (Okoro et al. 2010) Nouioui *et al*. 2018 as a later heterotypic synonym of *Lentzea atacamensis* (Okoro *et al*. 2010) Nouioui *et al*. 2018 and an emended description of *Lentzea atacamensis*

**Mo Ping[1], Zhao Yun-Lin[1]\*, Liu Jun[1], Gao Jian[2], Xu Zheng-Gang[1,3]\***

**1** Hunan Research Center of Engineering Technology for Utilization of Environmental and Resources Plant, Central South University of Forestry and Technology, Changsha, Hunan, China, **2** School of Life Science, Hunan University of Science and Technology, Xiangtan, Hunan, China, **3** College of Forestry, Northwest A & F University, Yangling, Shaanxi, China

\* zyl8291290@163.com (ZYL); xuzhenggang@nwafu.edu.cn (XZG)

## Abstract

The taxonomic relationship of *Lentzea atacamensis* and *Lentzea deserti* were re-evaluated using comparative genome analysis. The 16S rRNA gene sequence analysis indicated that the type strains of *L. atacamensis* and *L. deserti* shared 99.7% sequence similarity. The digital DNA-DNA hybridization (dDDH) and average nucleotide identity (ANI) values between the genomes of two type strains were 88.6% and 98.8%, respectively, greater than the two recognized thresholds values of 70% dDDH and 95–96% ANI for bacterial species delineation. These results suggested that *L. atacamensis* and *L. deserti* should share the same taxonomic position. And this conclusion was further supported by similar phenotypic and chemotaxonomic features between them. Therefore, we propose that *L. deserti* is a later heterotypic synonym of *L. atacamensis*.

## Introduction

The genus *Lentzea* was proposed by Yassin *et al*. in 1995 [1] and was subsequently emended by Lee *et al*. [2] and Fang *et al*. [3]. The strains of this genus form abundant aerial hyphae that fragment into rod-shaped elements. Whole-cell hydrolysates contain *meso*-diaminopimelic acid as diagnostic diamino acid and MK-9 (H$_4$) was the predominant menaquinone; The G+C content of genomic DNA ranges from 64.1 to 71.0 mol%. According to the List of LPSN (https://lpsn.dsmz.de/genus/lentzea), there are currently over 20 species of the genus *Lentzea* with validly published names. *Lentzea* species are distributed in different habitats, such as human pathological tissue [1], an equine placenta [4], a limestone [5] and different soils [6–8]. As separate genomic species, *Lechevalieria atacamensis* and *Lechevalieria deserti* were isolated from hyperarid soils of the Atacama Desert and proposed in 2010 as two validly named species based on a polyphasic taxonomic approach [9]. In 2018, *Lechevalieria atacamensis* and

**Data Availability Statement:** All relevant data are within the manuscript and its Supporting Information files.

**Funding:** This work was supported by the State Forestry and Grassland Bureau (2018-01 to ZYL), Major Science and Technology Program of Hunan Province (2017NK1014 to ZYL), Forestry Science and Technology Project of Hunan Province (XLK201825 to ZYL), Key Technology R&D Program of Changsha (kq190114 to ZYL) and Natural Science Foundation of Hunan Province (2019JJ50027 to XZG). The funders had no role in study design, data collection and analysis, decision to publish, or preparation of the manuscript.

**Competing interests:** The authors have declared that no competing interests exist.

*Lechevalieria deserti* were transferred to the genus *Lentzea* from the genus *Lechevalieria* based on comparative genome analysis [10]. However, our research showed that the two species belong to the same genome species. The aim of the present study was to clarify the relationship between the type strains of *L. atacamensis* and *L. deserti* based on genomic and associated phenotypic data.

## Materials and methods

### Phenotypic characterization

*L. atacamensis* CGMCC 4.5536[T] (= C61[T] = DSM 45479[T]) and *L. deserti* CGMCC 4.5535[T] (= C68[T] = DSM 45480[T]), were purchased from the CGMCC (China General Microbiological Culture Collection Center). The cultural properties of *L. atacamensis* CGMCC 4.5536[T] and *L. deserti* CGMCC 4.5535[T] were observed on various media including Gause's synthetic No. 1 medium [11] and ISP 2–7 media as described by Shirling and Gottlieb [12]. The color of aerial hyphae, substrate mycelia and soluble pigments were determined by the Color Standards and Color Nomenclature [13]. Growth at different temperature, pH range and NaCl tolerance was tested on ISP2 agar medium for 14 days according to the method described in the literature [12]. Carbon source utilization tests and enzyme activities were performed using the Biolog GEN III MicroPlates (Biolog, USA) and API ZYM system (bioMérieux, France) according to the manufacturer's instructions. Other biochemical tests such as hydrolysis of aesculin, starch and urea, hydrogen sulfide production and nitrate reduction, etc. were carried out according to the methods described by Xu *et al.* [14]. The isomer of diaminopimelic acid and sugar analysis of whole-cell hydrolysates were performed according to the procedures described by Hasegawa *et al.* [15] and Lechevalier and Lechevalier [16]. Menaquinones were extracted according to the method of Collins *et al.* [17] and analyzed by HPLC [18]. The experiments were carried out in triplicate. Whole genome sequences of the type strains of *L. atacamensis* and *L. deserti* are available from the GenBank database. Their genomic data such as GenBank assembly accessions, genomic lengths, number of contigs, contig N50 and DNA G+C contents, etc., are presented in Table 1.

**Table 1. Genomic data of *L. atacamensis* DSM 45479[T] (= CGMCC 4.5536[T] = C61[T]) and *L. deserti* DSM 45480[T] (= CGMCC 4.5535[T] = C68[T]).**

| Strain | *L. deserti* DSM 45480[T] | *L. atacamensis* DSM 45479[T] |
|---|---|---|
| Assembly accession | GCA_003148865.1 | GCA_003269295.1 |
| Genome size (bp) | 9,529,573 | 9,306,230 |
| No. of contigs | 50 | 38 |
| Contig N50 (bp) | 442,485 | 785,641 |
| G+C content (mol%) | 68.8 | 68.9 |
| Protein coding genes | 9221 | 9075 |
| RNA genes | 86 | 90 |
| rRNA genes | 8 | 12 |
| 5S rRNA | 6 | 6 |
| 16S rRNA | 1 | 5 |
| 23S rRNA | 1 | 1 |
| tRNA genes | 70 | 70 |
| Other RNA genes | 8 | 8 |
| Protein coding genes with function prediction | 6780 | 6635 |
| without function prediction | 2441 | 2440 |
| Protein coding genes with enzymes | 1589 | 1569 |

## Phylogenetic analysis and genomic DNA-DNA correlation analysis

The phylogenetic analysis of *L. atacamensis* C61[T] and *L. deserti* C68[T] was carried out using the Type (Strain) Genome Server (https://tygs.dsmz.de/) [19]. The average nucleotide identity (ANI) analysis was used for evaluating the genetic relationship between *L. atacamensis* C61[T] and *L. deserti* C68[T] by using the orthoANIu algorithm and an online ANI calculator (www.ezbiocloud.net/tools/ani) [20, 21]. At the same time, the digital DNA-DNA hybridization (dDDH) value between two genome sequences was calculated by the online Genome-to-Genome Distance Calculator (http://ggdc.dsmz.de/distcalc2.php) [22].

## Results and discussion

The phylogenomic analysis showed that *L. deserti* C68[T] was most closely related to *L. atacamensis* C61[T] (Fig 1), confirming that they should belong to the same genomic species. This result further suggested that the phylogenetic analysis based on whole genome sequences exhibited better resolution than the phylogenetic analysis based on 16S rRNA gene sequence [23]. ANI analysis indicated that *L. atacamensis* C61[T] and *L. deserti* C68[T] exhibited 98.8% ANI value, greater than the 95–96% threshold for species demarcation [24], confirming that they have the same taxonomic position. Meanwhile, *L. atacamensis* C61[T] and *L. deserti* C68[T] had an 88.6% dDDH value, greater than the 70% threshold for species demarcation, confirming that they belong to the same species [25]. In addition, comparative phenotypic characteristics

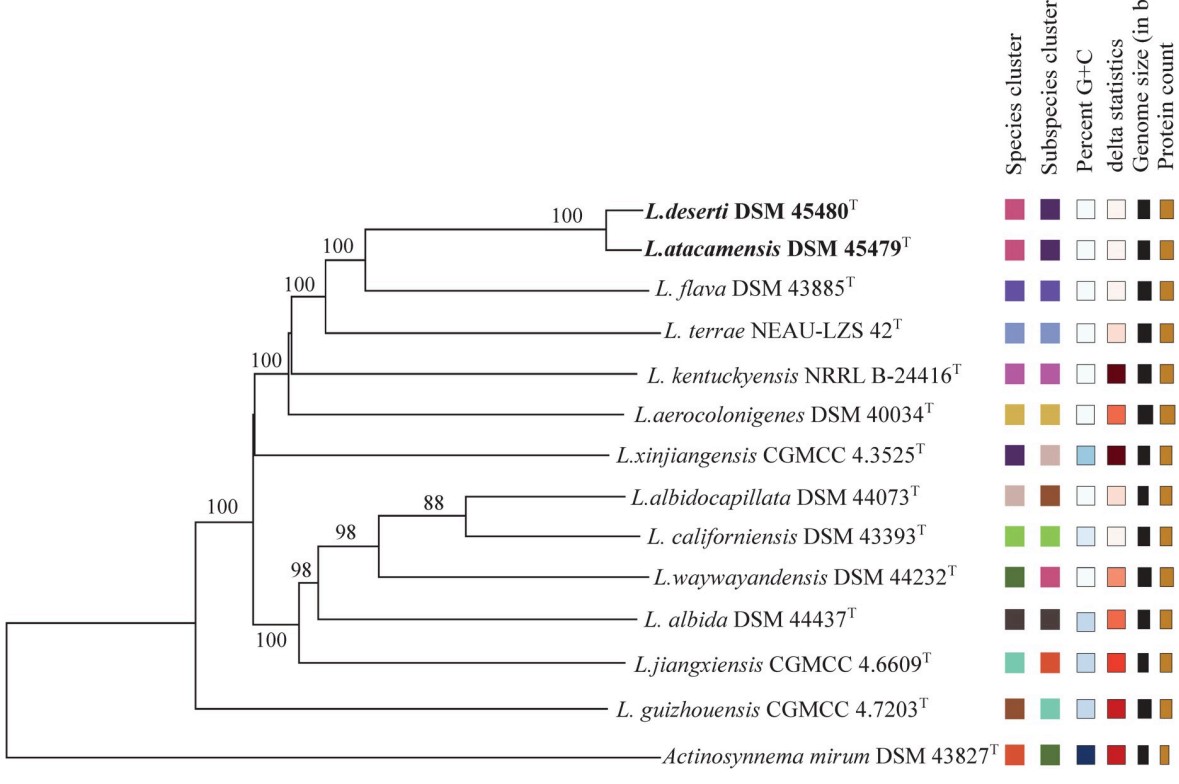

**Fig 1. Phylogenetic tree based on whole genome sequences of *L. atacamensis* DSM 45479[T], *L. deserti* DSM 45480[T] and related reference strains.** Tree inferred with FastME 2.1.6.1 [27] from GBDP distances calculated from genome sequences. The branch lengths are scaled in terms of GBDP distance formula d5. The numbers above branches are GBDP pseudo-bootstrap support values >60% from 100 replications, with an average branch support of 96.0%. The tree was rooted at the midpoint [28].

of *L. atacamensis* CGMCC 4.5536[T] and *L. deserti* CGMCC 4.5535[T] are presented in Table 2 and S1 Fig. As shown in Table 1 and S1 Fig, most features between them were almost identical. For example, they were produced white aerial mycelium on ISP 3, ISP 4 and ISP 5. In Biolog GENIII test, positive for growth on acetoacetic acid, acetic acid, acid Methyl Ester, aztreonam, etc.; And negative for growth on D-aspartic acid, D-fructose-6-PO$_4$, D-fucose etc. They both contained *meso*-DAP in the cell-wall, and galactose, mannose and rhamnose as the whole-cell sugar. The major menaquinones of strains consisted of MK-9 (H$_4$) (Table 1). At the same time, the genomes size of strain *L. atacamensis* DSM 45479[T] is 9,306,230 bp in 38 of contigs. 9075 protein coding genes, 90 rRNA genes, 6635 protein coding genes, 2440 without function prediction and 1569 protein coding genes with enzymes were predicted. While the genomes size of strain *L. deserti* DSM 45480[T] is 9,529,573 bp in 50 of contigs. 9221 protein coding genes, 86 rRNA genes, 6780 protein coding genes, 2442 without function prediction and 1589 protein coding genes with enzymes were predicted (Table 2). Consequently, we propose that *L. deserti* C68[T] is a later heterotypic synonym of *L. atacamensis* C61[T] based on the results above and rule 42 of the Bacteriological Code [26].

**Table 2. Phenotypic characteristics between *L. atacamensis* CGMCC 4.5536[T] and *L. deserti* CGMCC 4.5535[T].**

| Characteristic | 1 | 2 |
|---|---|---|
| Aerial mycelium on Gause's synthetic No. 1 | TB | TB |
| Aerial mycelium on ISP 2 | AY | WAAY |
| Aerial mycelium on ISP 3, 4 and 5 | WH | WH |
| Substrate mycelium on ISP 2, 3, 4 and 5 | YO | YO |
| Melanin or diffusible pigments on ISP 6 and 7 | – | – |
| [a]Acid production from: | | |
| Adonitol, erythritol, melezitose, methyl, $\alpha$-D-glucoside | – | + |
| Inulin, lactose, mannitol, mannose, rhamnose, salicin, trehalose | + | + |
| API ZYM test | | |
| Acid phosphatase, alkaline phosphatase, cystine arylamidase, esterase (C4) | + | + |
| Esterase lipase (C8), trypsin, leucine arylamidase, lipase (C14) | + | + |
| *N*-acetyl-$\beta$-glucosaminidase, valine arylamidase, $\alpha$-chymotrypsin, $\alpha$-mannosidase | + | + |
| $\alpha$-galactosidase, $\alpha$-glucosidase, $\beta$-galactosidase, $\beta$-glucosidase | + | + |
| naphtol-AS-BI-phosphohydrolase, $\beta$-glucuronidase | W | W |
| $\alpha$-fucosidase | – | – |
| BiologGEN III test | | |
| Acetoacetic acid, acetic acid, aztreonam, bromo-succinic acid, D-arabitol, D-cellobiose, dextrin, D-fructose, D-galactose, D-gluconic acid, D-glucuronic acid, D-lactic acid, D-maltose, D-mannitol, D-mannose, D-melibiose, D-malic acid, D-raffinose, D-salicin, D-serine, D-trehalose, D-turanose, gentiobiose, gelatin, glycerol, glycyl-L-proline, inosine, lactate, L-fucose, *myo*-inositol, L-alanine, L-arginine, L-glutamic acid, L-malic acid, L-serine, L-histidine, lithium chloride, L-pyroglutamic acid, L-rhamnose, L-aspartic acid, methyl pyruvate, *N*-acetyl-D-glucosamine, nalidixic acid, propionic acid, sodium bromate, sucrose, stachyose, $\alpha$-D-glucose, $\alpha$-D-lactose, 1% sodium citric acid, Tween 40, $\alpha$-keto-glutaric acid, $\gamma$-amino-butryric acid, $\beta$-hydroxy-D, L-butyric acid. | + | + |
| D-aspartic acid, D-fructose-6-PO$_4$, D-fucose, D-serine, D-sorbitol, formic acid, L-galactonic acid lactone, L-lactic acid, minocycline, *N*-acetyl-$\beta$-D-mannosamine, *N*-acetyl-D-galactosamine, *N*-acetyl neuraminic acid, 3-methyl glucose, fusidic acid D-galacturonic acid, D-sorbitol, D-glucose-6-PO$_4$, guanidine HCl, niaproof 4, quinic acid, D-saccharic acid, sodium butyrate, tetrazolium violet, tetrazolium blue, troleandomycin, *p*-hydroxy-phenylacetic acid, $\alpha$-hydroxy-butyric acid, $\beta$-methyl-D-glucoside | – | – |
| Lincomycin | + | – |
| Acid Methyl Ester, pectin | W | W |

(*Continued*)

**Table 2.** (Continued)

| Characteristic | 1 | 2 |
|---|---|---|
| Glucuronamide, $\alpha$-keto-butyric acid | + | W |
| Mucic acid, vancomycin | W | – |
| Potassium tellurite | W | + |
| Hydrolysis of starch, aesculin and allantoin | + | + |
| $H_2S$ production, nitrate reduction and urea hydrolysis | – | – |
| [a] Decomposition of: arbutin, uric acid | – | + |
| [a] Decomposition of: casein, elastin, hypoxanthine, tyrosine, xylan | + | + |
| Growth at/in: 4% NaCl (w/v), 20–40°C and pH 5.0–11.0 | + | + |
| Growth at/in: 0.1% Methyl violet (w/v) | – | – |
| [a] Major fatty acids (%) | | |
| $iso$-$C_{16:0}$ | 23.1 | 24.5 |
| $C_{16:0}$ | 9.9 | 10.8 |
| $anteiso$-$C_{15:0}$ | 9.3 | 9.7 |
| $anteiso$-$C_{17:0}$ | 8.5 | 8.4 |
| $iso$-$C_{15:0}$ | 6.6 | 7.5 |
| $C_{16:1}$ | 6.5 | 5.8 |
| $iso$-$C_{14:0}$ | 3.5 | 3.5 |
| $iso$-$C_{17:0}$ | 1.8 | 2.8 |
| $C_{17:0}$ | 1.5 | ND |
| Predominant menaquinones | MK-9 $(H_4)$ | MK-9 $(H_4)$ |
| Cell-wall diamino acid | $mes$-DAP | $mes$-DAP |
| Whole-cell sugars | G, M, R | G, M, R |

Note:

a, data are from Okoro *et al.* [9].

Note: 1, *L. atacamensis* CGMCC 4.5536[T]; 2, *L. deserti* CGMCC 4.5535[T]. All data are from this study except where specified; +, Positive reaction;–, negative reaction; W, weak reaction; ND, Not detected; TB, Tilleul-buff; AY, Antimony yellow; YO, Yellow ocher; WAAY, White and Antimony yellow; WH, White; G, galactose; M, mannose; R, rhamnose.

## Emended description of *Lentzea atacamensis* (Okoro *et al.* 2010) Nouioui *et al.* 2018

The description is as before (Okoro *et al.* 2010 and Nouioui *et al.* 2018) with the following additions. Tilleul-buff aerial mycelia and olive-buff massicot yellow substrate mycelia are produced on Gause's synthetic agar medium at 28°C for 21 days. Positive for acid phosphatase, alkaline phosphatase, cystine arylamidase, esterase (C4), esterase lipase (C8), trypsin, leucine arylamidase, lipase (C14), *N*-acetyl-$\beta$-glucosaminidase, naphtol-AS-BI-phosphohydrolase, valine arylamidase, $\alpha$-chymotrypsin, $\alpha$-mannosidase, $\alpha$-galactosidase, $\alpha$-glucosidase, $\beta$-galactosidase, $\beta$-glucosidase$\beta$-glucuronidase; but negative for $\alpha$-fucosidase. The cell wall contains *meso*-DAP. Whole-cell sugars are galactose, mannose and rhamnose. The predominant menaquinones is MK-9 $(H_4)$. The DNA G+C content of the genome sequence, consisting of 19306230 bp, is 68.9 mol%.

The type strain is CGMCC 4.5536 (= C61 = NRRL B-24706 = JCM 17492 = DSM 45479).

## Supporting information

**S1 Fig. Cultural characteristics of strains in different medium after 21d of incubation at 28˚C.**
(DOCX)

## Acknowledgments

We sincerely thank MCCC (Marine Culture Collection of China) for providing excellent technical assistance.

## Author Contributions

**Data curation:** Zhao Yun-Lin, Gao Jian, Xu Zheng-Gang.

**Formal analysis:** Gao Jian, Xu Zheng-Gang.

**Investigation:** Mo Ping, Liu Jun.

**Software:** Zhao Yun-Lin, Gao Jian, Xu Zheng-Gang.

**Writing – original draft:** Mo Ping, Liu Jun, Gao Jian, Xu Zheng-Gang.

**Writing – review & editing:** Mo Ping, Zhao Yun-Lin, Gao Jian, Xu Zheng-Gang.

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
