## [Decision Letter · Decision Letter 0]

2 Oct 2020

PONE-D-20-29208

Proposal of Lentzea deserti (Okoro et al. 2010) Nouioui et al. 2018 as a later heterotypic synonym of Lentzea atacamensis (Okoro et al. 2010) Nouioui et al. 2018

PLOS ONE

Dear Dr. Zhenggang,

Thank you for submitting your manuscript to PLOS ONE. After careful consideration, we feel that it has merit but does not fully meet PLOS ONE’s publication criteria as it currently stands. Therefore, we invite you to submit a revised version of the manuscript that addresses the points raised during the review process.

We look forward to receiving your revised manuscript.

Kind regards,

Yogesh S. Shouche

Academic Editor

PLOS ONE

Journal Requirements:

2.Thank you for stating the following in the Funding Information Section of your manuscript:

[Major Science and Technology Program of Hunan Province, Grant/Award Number:

26 2017NK1014; Forestry Science and Technology Project of Hunan Province, Grant/Award

27 Number: XLK201825; Key Technology R&D Program of Changsha, Grant/Award Number:

28 kq190114; State Forestry and Grassland Bureau, Grant/Award Number: 2018-01.]

 [No]

3.Thank you for stating the following in your Competing Interests section: 

[No].

Reviewers' comments:

Reviewer's Responses to Questions

**Comments to the Author**

1. Is the manuscript technically sound, and do the data support the conclusions?

Reviewer #1: Yes

Reviewer #2: Yes

2. Has the statistical analysis been performed appropriately and rigorously? 

Reviewer #1: Yes

Reviewer #2: N/A

3. Have the authors made all data underlying the findings in their manuscript fully available?

Reviewer #1: Yes

Reviewer #2: Yes

4. Is the manuscript presented in an intelligible fashion and written in standard English?

Reviewer #1: Yes

Reviewer #2: Yes

5. Review Comments to the Author

Reviewer #1: Background

In a genomic-based classification of the type strains of Lentzea atacamensis and Lentzea deserti the authors have correctly concluded that the organisms belong to a single species. Nomenclatural matters can cause confusion at the best of times hence it is important that the authors present their conclusions in an unambiguous way and in so doing address the points raised below.

Major points

1. In the paper by Okoro et al. (2010) Lechevalieria atacamensus has priority over Lechavalieria deserti hence the latter needs to be proposed as a later heterotypic synonym of the former.

2. The title of the paper needs to be amended by adding that an "emended description of Lentzea atacamensis is given."

3. It needs to be made clear, especially in the Abstract and Introduction, that the proposal to recognise Lentzea deserti as a later hetertoypic synonym of Lentzea atacamensis is based on phenotypic as well as genomic data even though the latter outweigh the former.

4. The Introduction needs to be revised along the following lines:

(a) The initial classification of the genera Lechevalieria and Lentzea and the perceived relationship between them reflected the limited resolution of the taxonomic methods available at the time. Please add appropriate references.

(b) The application of phylogenomic and associated phenotypic methods clearly showed that Lechevaleria should be seen as a subjective synonym of Lentzea.

(c) The genus Lentzea currently includes 20 validly named species including Lentzea atacamensis and Lentzea deserti which were previously classified as Lechevalieria atacamensis and Lechevalieria deserti. Please add appropriate references.

(d) The aim of the present study was to clarify the relationship between the type strains of L. atacamensis and L. deserti based on genomic and associated phenotypic data.

(e) Please note that the genus Lentzea has been emended by Labeda et al. (2001) and later by Fang et al. (2017. IJSEM 67 : 2357-2362). In contrast, it has not been emended by Lee et al. (2000).

(f) The authors note the anomalous position of "Lechevalieria rhizosphaerae" but have not addressed this issue. Their paper would be much improved were they to do so.

Minor points

Line 45. The term "first" is redundant.

Line 60. Lentzea atacamensis ....Delete "The type strain".

Line 68. ISP2 agar.

Line 86. Reference 14 not 11.

Lines 88-92. Might be wise here and get straight to the point and say that the results of this study provide further evidence that data derived from draft whole-genome sequences provide much greater resolution than 16S rRNA gene sequence studies (plus references) and then go on to consider the hard data.

Lines 99-102. The phenotypic data need to be given a little more emphasis as they underpin the genomic results. Were the results of the triplicate tests identical?

Lines 107-130.

* Line 110. "additions"

* The description should be based on the results obtained on each of the type strains.

* It should also include data on genome size and digital DNA G+C ratios.

References.

* Apart from the first word in the titles all other words should be lower case.

* Line 172. "Systematics".

Line 21. Phylogenomic tree is more precise than phylogenetic tree.

Figure 2. The plates are really nice but these results are probably best presented as supplementary data.

Reviewer #2: The authors proposed Lentzea deserti as a later heterotypic synonym of Lentzea atacamensis based on phenotypic, phylogenetic and genomic data. The content is well managed but the authors should reconsider the following points:

The introduction section is poor and should be improved by providing a proper taxonomic description of the taxa and their features (habitat, chemotaxonomic, genetic traits etc)

The phylogenetic relationships between Lentzea deserti and Lentzea atacamensis as well as with their close neighbours should be discussed and compared with previous studies. More genomic features should be added and discussed in the text and not just refer to Table 1. The morphology of strains 45480 and 45479 should be commented in the text.

No chemotaxonomic analyses were carried out in this present report. It would be good if the authors could add to this emended species some additional chemotaxonomic traits such as polar lipids, quinone and sugars (optional). The Fatty acids analysis should be performed for both strains in the same time and under the same condition. Since the genomes of both strains are available, it would improve the manuscript if the authors could highlight some interesting features of this proposed emended species (optional).

In the emendation section, Biolog section should be in the text and not in the emendation section. The authors should also include the genome sizes and the G+C content for the species L. atacamensis.

6. PLOS authors have the option to publish the peer review history of their article (what does this mean?). If published, this will include your full peer review and any attached files.

Reviewer #1: **Yes: **Professor Michael Goodfellow

Reviewer #2: No

---

## [Author Response · Author response to Decision Letter 0]

3 Nov 2020

Dear Yogesh S. Shouche and reviews,

Thank you for sending me the reviewers’ comments. Taking the comments into consideration, we have thoroughly revised the manuscript and re-submit the revised manuscript together with the responses to relevant comments. All changes were marked in the manuscript and have been approved by all authors.

We hope that the revision is adequate and thank you for considering our manuscript.

Please contact with us if any other necessary.

Best regards,

Mo Ping and Xu Zhenggang

E-mail: xuzhenggang@nwafu.edu.cn

Comments from Reviewer 1

1. In the paper by Okoro et al. (2010) Lechevalieria atacamensus has priority over Lechavalieria deserti hence the latter needs to be proposed as a later heterotypic synonym of the former.

Response: Thinks! We have made changes according to your comment.

2. The title of the paper needs to be amended by adding that an "emended description of Lentzea atacamensis is given."

Response: Thanks for the comment. Revisions have been made accordingly (please see Line 4-7). We are grateful for your this comment. We accept your suggestion and changed the title as “Proposal of Lentzea deserti (Okoro et al. 2010) Nouioui et al. 2018 as a later heterotypic synonym of Lentzea atacamensis (Okoro et al. 2010) Nouioui et al. 2018 and an emended description of Lentzea atacamensis”.

3. It needs to be made clear, especially in the Abstract and Introduction, that the proposal to recognise Lentzea deserti as a later hetertoypic synonym of Lentzea atacamensis is based on phenotypic as well as genomic data even though the latter outweigh the former. 

Response: Thanks for the comment. Revisions have been made accordingly (please see Line 33-36 and Line 46-48).

In abstract, we revised: These results suggested that L. atacamensis and L. deserti should share the same taxonomic position. And this conclusion was further supported by similar phenotypic and chemotaxonomic features between them (Line 33-36).

In introduction, we revised: According to the List of LPSN (https://lpsn.dsmz.de/genus/lentzea), there are currently over 20 species of the genus Lentzea with validly published names [9] (Line 46-48).

4. The Introduction needs to be revised along the following lines:

(a) The initial classification of the genera Lechevalieria and Lentzea and the perceived relationship between them reflected the limited resolution of the taxonomic methods available at the time. Please add appropriate references.

Response: Thanks for the comment. Revisions have been made accordingly (please see Line 50-53).

We revised: As separate genomic species, Lechevalieria atacamensis and Lechevalieria deserti were isolated from hyperarid soils of the Atacama Desert and proposed in 2010 as two validly named species based on a polyphasic taxonomic approach [9] (Line 50-53).

In the current systematics, the classification of prokaryote is based on a so-called polyphasic taxonomic approach, comprised of phenotypic, chemotaxonomic and genotypic data, as well as phylogenetic information. Of these, classical DNA–DNA hybridization (DDH) plays a key role in determining procaryotic species relatedness as it provides a clear and objective numerical threshold for a species boundary, for which 70% DDH was suggested. DDH has been the ‘gold standard’ for bacterial species demarcation over the last 50 years, but DDH procedures are known to be labor-intensive, error-prone and do not allow the generation of cumulative databases. Thus, there has been an urgent need for an alternative genotype-based standard. With the development of whole genome sequencing technology, many efforts have been made to develop a bioinformatic method to replace DDH for differentiating species. These efforts were mainly focused on devising values analogous to DDH values, such as genome BLAST distance phylogeny (GBDP), average nucleotide identity (ANI) and maximal unique matches index (MUMi). Of these, ANI has been most widely used as a possible next-generation gold standard for species delineation. At present, the proposed and generally accepted species boundary for ANI value is 95–96%, equate to a DDH value of 70%.

In Okoro's study, phenotypic characterization, chemotaxonomic characterization and phylogenetic analysis of 16S rRNA gene sequence are not sufficient to categorize L. deserti as distinct from L. atacamensis; And they were proposed to be two novel species, it is mainly because 41.8% classic DNA-DNA hybridization value between the type strains of L. deserti and L. atacamensis was well below the threshold used to delineate prokaryote species. Actually, Okoro CK et al. found that most features between them were almost identical, such as phenotypic properties, spore morphology and fatty acids.However, in our study, phenotypic characterization, chemotaxonomic characterization, genome phylogenetic analysis and digit DNA-DNA correlation analysis between L. atacamensis CGMCC 4.5536T (=C61T=DSM 45479T) and L. deserti CGMCC 4.5535T (=C68T=DSM 45480T) are sufficient to prove that they belong to the same species.

(b) The application of phylogenomic and associated phenotypic methods clearly showed that Lechevaleria should be seen as a subjective synonym of Lentzea.

Response: Thanks for the comment. Revisions have been made accordingly (please see Line 104-117).

We revised: As shown in Table 1 and S1 Fig, most features between them were almost identical. For example, they were produced white aerial mycelium on ISP 3, ISP 4 and ISP 5. In Biolog GENIII test, positive for growth on acetoacetic acid, acetic acid, acid Methyl Ester, aztreonam, etc.; And negative for growth on D-aspartic acid, D-fructose-6-PO4, D-fucose etc. They both contained meso-DAP in the cell-wall, and galactose, mannose and rhamnose as the whole-cell sugar. The major menaquinones of strains consisted of MK-9 (H4) (Table 1). At the same time, the genomes size of strain L. atacamensis DSM 45479T is 9,306,230 bp in 38 of contigs. 9075 protein coding genes, 90 rRNA gene, 6635 protein coding genes, 2440 without function prediction and 1569 protein coding genes with enzymes were predicted. While the genomes size of strain L. deserti DSM 45480T is 9,529,573 bp in 50 of contigs. 9221 protein coding genes, 86 rRNA gene, 6780 protein coding genes, 2442 without function prediction and 1589 protein coding genes with enzymes were predicted (Table 2).

(c) The genus Lentzea currently includes 20 validly named species including Lentzea atacamensis and Lentzea deserti which were previously classified as Lechevalieria atacamensis and Lechevalieria deserti. Please add appropriate references.

Response: Thanks for the comment. Revisions have been made accordingly (please see Line 46-48).

(d) The aim of the present study was to clarify the relationship between the type strains of L. atacamensis and L. deserti based on genomic and associated phenotypic data.

Response: Thanks for the comment. Revisions have been made accordingly (please see Question and Reply 4b).

(e) Please note that the genus Lentzea has been emended by Labeda et al. (2001) and later by Fang et al. (2017. IJSEM 67: 2357-2362). In contrast, it has not been emended by Lee et al. (2000).

Response: Thanks for the comment. Revisions have been made accordingly (please see Line 42).

(f) The authors note the anomalous position of "Lechevalieria rhizosphaerae" but have not addressed this issue. Their paper would be much improved were they to do so.

Response: Revised. Thanks for the comment.

Minor point

5. Line 45. The term "first" is redundant. 

Response: Revised.

6. Line 60. Lentzea atacamensis ....Delete "The type strain". 

Response: Revised.

7. Line 68. ISP2 agar.

Response: Revised (please see Line 69).

8. Line 86. Reference 14 not 11.

Response: Revised.

9. Lines 88-92. Might be wise here and get straight to the point and say that the results of this study provide further evidence that data derived from draft whole-genome sequences provide much greater resolution than 16S rRNA gene sequence studies (plus references) and then go on to consider the hard data.

Response: Thanks for the comment. Revisions have been made accordingly (please see Line 93-97). 

The phylogenomic analysis showed that L. deserti C68T was most closely related to L. atacamensis C61T (Fig. 1), confirming that they should belong to the same genomic species. This result further suggested that the phylogenetic analysis based on whole genome sequences exhibited better resolution than the phylogenetic analysis based on 16S rRNA gene sequence [23].

10. Lines 99-102. The phenotypic data need to be given a little more emphasis as they underpin the genomic results. Were the results of the triplicate tests identical?

Response: Thanks for the comment. Revisions have been made accordingly (please see Line 75-78, Line 108-110, Line 129-132 and Line 307).

We have added experiment: The isomer of diaminopimelic acid and sugar analysis of whole-cell hydrolysates were performed according to the procedures described by Hasegawa et al. [15] and Lechevalier and Lechevalier [16]. Menaquinones were extracted according to the method of Collins et al. [17] and analyzed by HPLC [18].

Yes, the results of the triplicate tests are same.

11. Lines 107-130.

* Line 110. "additions"

Response: Revised (please see Line 123) .

* The description should be based on the results obtained on each of the type strains.

* It should also include data on genome size and digital DNA G+C ratios. 

Response: Thanks for the comment. Revisions have been made accordingly (please see Line110-117and Line 122-132).

12. References.

* Apart from the first word in the titles all other words should be lower case.

* Line 172. "Systematics".

Response: Revised (please see Line 195-196 etc.).

13. Line 21. Phylogenomic tree is more precise than phylogenetic tree.

Thinks, I agree with you (please see Line 93-96).

14. Figure 2. The plates are really nice but these results are probably best presented as supplementary data.

Thanks for the comment. Revisions have been made accordingly (please see Line 138-140 and Supporting file).

Comments from Reviewer 2 

The authors proposed Lentzea deserti as a later heterotypic synonym of Lentzea atacamensis based on phenotypic, phylogenetic and genomic data. The content is well managed but the authors should reconsider the following points:

15. The introduction section is poor and should be improved by providing a proper taxonomic description of the taxa and their features (habitat, chemotaxonomic, genetic traits etc

Thanks for the comment. Revisions have been made accordingly (please see Line 42-50).

The strains of this genus form abundant aerial hyphae that fragment into rod-shaped elements. Whole-cell hydrolysates contain meso-diaminopimelic acid as diagnostic diamino acid and MK-9 (H4) was the predominant menaquinone; The G+C content of genomic DNA ranges from 64.1 to 71.0 mol%. According to the List of LPSN (https://lpsn.dsmz.de/genus/lentzea), there are currently over 20 species of the genus Lentzea with validly published names. Lentzea species are distributed in different habitats, such as human pathological tissue [1], an equine placenta [4], a limestone [5] and different soils [6-8].

16. The phylogenetic relationships between Lentzea deserti and Lentzea atacamensis as well as with their close neighbors should be discussed and compared with previous studies. More genomic features should be added and discussed in the text and not just refer to Table 1. The morphology of strains 45480 and 45479 should be commented in the text.

Thanks for the comment. Revisions have been made accordingly (please see Line 50-54 and Line 102-117).

We added: In addition, comparative phenotypic characteristics of L. atacamensis CGMCC 4.5536T and L. deserti CGMCC 4.5535T are presented in Table 2 and S1 Fig. As shown in Table 1 and S1 Fig, most features between them were almost identical. For example, they were produced white aerial mycelium on ISP 3, ISP 4 and ISP 5. In Biolog GENIII test, positive for growth on acetoacetic acid, acetic acid, acid Methyl Ester, aztreonam, etc.; And negative for growth on D-aspartic acid, D-fructose-6-PO4, D-fucose etc. They both contained meso-DAP in the cell-wall, and galactose, mannose and rhamnose as the whole-cell sugar. The major menaquinones of strains consisted of MK-9 (H4) (Table 1). At the same time, the genomes size of strain L. atacamensis DSM 45479T is 9,306,230 bp in 38 of contigs. 9075 protein coding genes, 90 rRNA gene, 6635 protein coding genes, 2440 without function prediction and 1569 protein coding genes with enzymes were predicted. While the genomes size of strain L. deserti DSM 45480T is 9,529,573 bp in 50 of contigs. 9221 protein coding genes, 86 rRNA gene, 6780 protein coding genes, 2442 without function prediction and 1589 protein coding genes with enzymes were predicted (Table 2).

17. No chemotaxonomic analyses were carried out in this present report. It would be good if the authors could add to this emended species some additional chemotaxonomic traits such as polar lipids, quinone and sugars (optional). 

Thanks for the comment. Revisions have been made accordingly (please see Line 75-78). 

We have added the additional chemotaxonomic traits under the same conditions: the isomer of diaminopimelic acid analysis and sugar analysis of whole-cell hydrolysates, menaquinones.

18. The Fatty acids analysis should be performed for both strains in the same time and under the same condition.

Thanks for the comment. We have not repeated the fatty acids, because the fatty acids of strains L. atacamensis and L. deserti were from the same paper (Okoro et al. [9]). They should compare in the same time and under the same condition. 

19. Since the genomes of both strains are available, it would improve the manuscript if the authors could highlight some interesting features of this proposed emended species (optional).

In the further, we will carry out some interesting features of strains.

20. In the emendation section, Biolog section should be in the text and not in the emendation section. 

Thanks for the comment. Revisions have been made accordingly (please see Line 106-108).

21. The authors should also include the genome sizes and the G+C content for the species L. atacamensis.

Thanks for the comment. Revisions have been made accordingly (please see Line 106-108).. We have added the genome sizes and the G+C content for the species L. atacamensis: The DNA G+C content of the genome sequence, consisting of 19,306,230 bp, was 68.9 mol%.

---

## [Decision Letter · Decision Letter 1]

21 Jan 2021

Proposal of Lentzea deserti (Okoro et al. 2010) Nouioui et al. 2018 as a later heterotypic synonym of Lentzea atacamensis (Okoro et al. 2010) Nouioui et al. 2018 and an emended description of Lentzea atacamensis

PONE-D-20-29208R1

Dear Dr. Xu,

We’re pleased to inform you that your manuscript has been judged scientifically suitable for publication and will be formally accepted for publication once it meets all outstanding technical requirements.

Kind regards,

Feng ZHANG, Ph.D.

Academic Editor

PLOS ONE

---

## [Editor Report · Acceptance letter]

26 Jan 2021

PONE-D-20-29208R1 

Proposal of *Lentzea deserti* (Okoro et al. 2010) Nouioui et al. 2018 as a later heterotypic synonym of *Lentzea atacamensis* (Okoro et al. 2010) Nouioui et al. 2018 and an emended description of *Lentzea atacamensis*

Dear Dr. Zheng-Gang:

I'm pleased to inform you that your manuscript has been deemed suitable for publication in PLOS ONE. Congratulations! Your manuscript is now with our production department. 

Kind regards, 

on behalf of

Dr. Feng ZHANG 

Academic Editor

PLOS ONE